# In Vitro Influenza A Virus-Inactivating Activity of HIDROX^®^, Hydroxytyrosol-Rich Aqueous Olive Pulp Extract

**DOI:** 10.3390/pathogens14060529

**Published:** 2025-05-25

**Authors:** Mayar Yasser Zeinelabideen Mohamed, Dulamjav Jamsransuren, Sachiko Matsuda, Koichi Narita, Toshihiro Murata, Haruko Ogawa, Yohei Takeda

**Affiliations:** 1Department of Veterinary Medicine, Obihiro University of Agriculture and Veterinary Medicine, 2-11 Inada, Obihiro 080-8555, Japan; mayaryasser0@gmail.com (M.Y.Z.M.); hogawa@obihiro.ac.jp (H.O.); 2Research Center for Global Agromedicine, Obihiro University of Agriculture and Veterinary Medicine, 2-11 Inada, Obihiro 080-8555, Japan; jduuya@obihiro.ac.jp (D.J.); chakachaka0810@gmail.com (S.M.); 3Faculty of Pharmaceutical Sciences, Tohoku Medical and Pharmaceutical University, 4-4-1 Komatsushima, Aoba-ku, Sendai 981-8558, Japan; k-narita@tohoku-mpu.ac.jp (K.N.); murata-t@tohoku-mpu.ac.jp (T.M.)

**Keywords:** aqueous olive pulp extract, hydroxytyrosol, influenza A virus, virucidal activity

## Abstract

Influenza A virus (IAV) is an important respiratory pathogen. We evaluated the IAV-inactivation activity of hydroxytyrosol (HT)-rich aqueous olive pulp extract (HIDROX^®^) and its mechanisms. The HIDROX-containing solution and cream showed concentration- and time-dependent virucidal activity. The virucidal activity of HIDROX was higher than pure HT. With Western blotting (WB), the band intensities of multiple viral structural proteins in HIDROX- and HT-treated viruses were weaker than in the control, and high-molecular-mass bands were observed. These results suggest that HIDROX and HT may have induced the structural changes or abnormalities of viral proteins. HIDROX and HT had no or limited impact on hemagglutination and neuraminidase activities, as well as the virus genome. No apparent abnormalities in the viral particles were observed through electron microscopy following treatment with HIDROX and HT. Treatment with HT, but not HIDROX, resulted in the production of high levels of reactive oxygen species (ROS) and/or reactive nitrogen species (RNS). In HT treatment but not HIDROX treatment, the virucidal activity disappeared, and the induction of abnormal band patterns of a viral protein in WB was cancelled by ROS/RNS scavenger activity. These findings showed the possible utility of HIDROX as a naturally derived IAV virucidal component that may contribute to IAV control.

## 1. Introduction

Influenza A virus (IAV) is a contagious virus that causes respiratory disease and is characterized by a high mutation rate and wide distribution worldwide. Humans can be infected with both IAV and influenza B virus, which cause annual influenza epidemics producing significant morbidity and mortality globally [1]. The Centers for Disease Control and Prevention estimated that there were 36 million influenza illnesses, 18 million influenza-associated medical visits, 400,000 influenza-related hospitalizations, and 22,000 influenza-associated deaths in the USA between 2019 to 2020 [2]. Although an influenza vaccine is available, complete protection cannot be achieved by the vaccination alone because of low annual vaccine coverage, limited immunostimulatory action in older adults, and the occasional occurrence of antigenic mismatch between the vaccine and currently circulating strains [3]. Therefore, both vaccination and treatment using anti-influenza viral drugs are required for stricter control of influenza. Currently, multiple therapeutic agents, such as neuraminidase (NA) inhibitors (e.g., oseltamivir and zanamivir) and matrix-2 (M2) ion channel blockers (e.g., amantadine and rimantadine), as well as polymerase acidic (PA) endonuclease inhibitors (e.g., baloxavir marboxil), have been developed [3]. However, IAV strains resistant to NA inhibitors and M2 ion channel inhibitors have emerged [4,5,6]. In addition, the emergence of variant strains with amino acid substitutions that may reduce the therapeutic efficacy of a PA endonuclease inhibitor have been reported [7]. Given the current circumstances, the use of virucidal disinfectants to maintain hand hygiene and inactivate viruses in environments to prevent IAV transmission is also important; these measures may complement vaccines and antiviral treatments, thereby reducing the risk of IAV circulation.

In addition to the currently available chemical disinfectants, naturally derived components have been intensively studied in recent years for their potential application as harmless and eco-friendly virucidal agents [8]. Plant polyphenols have virucidal and antiviral effects against pathogenic viruses [9,10]. Hydroxytyrosol (HT) is one of the main simple phenols contained in olive (*Olea europaea* L.) extract and oil. The molecular weight of HT is 154, and the chemical formula is C_8_H_10_O_3_. Pure HT is a clear, tasteless, colorless liquid that can be either hydrosoluble or liposoluble [11]. HT was reported to have virucidal activity against enveloped viruses such as H1N1, H3N2, H5N1, and H9N2 subtype IAVs and Newcastle disease virus but not against nonenveloped viruses such as bovine rotavirus and fowl adenovirus [12]. In addition, the antiviral activity of HT against human immunodeficiency virus (HIV) was reported [13]. HT, tyrosol, oleuropein, oleocanthal, and oleacein are the primary phenolic compounds found in olive oil and have a wide range of biological effects, including antiatherogenic, antioxidant, and anti-inflammatory activities [14,15]. HIDROX^®^ (Oliphenol LLC., Hayward, CA, USA) is a standardized freeze-dried powder of aqueous extract made from a byproduct of the oil extraction process [11]. The main biological constituents of HIDROX are olive-derived polyphenols, accounting for ~12% of all components in HIDROX. HT is especially abundant in HIDROX and accounts for ~40% of total polyphenols (~5% of all components in HIDROX) [16]. A HIDROX-containing bilayer film has been reported to alleviate synovitis and inflammation in a rat arthritis model [17]. In addition, HIDROX counteracts the neurodegenerative processes in a mouse model of Parkinson’s disease [18] and was reported to be low-toxic in vitro and in vivo [19], implying that HIDROX is suitable for human use. As a matter of fact, HIDROX is certified as Generally Recognized as Safe by the American Food and Drug Administration. Recently, HIDROX and HT were demonstrated to have virus-inactivating activity in solution and cream forms against severe acute respiratory syndrome coronavirus 2 (SARS-CoV-2). In tests of the solution formulation, the virucidal activities of 0.90 mg/mL HIDROX were compared with those of 0.90 mg/mL and 0.05 mg/mL HT solutions. The HT concentration in 0.90 mg/mL HIDROX is estimated to be 0.05 mg/mL. In this analysis, the 0.90 mg/mL HIDROX solution exhibited stronger SARS-CoV-2 inactivation activity than both the 0.90 mg/mL and 0.05 mg/mL pure HT solutions [16]. Since HT has already demonstrated virucidal activity against IAV [12], we investigated whether HIDROX showed more potent IAV-inactivating activity than pure HT. Thus, this study evaluated the virucidal activity of HIDROX against IAV and elucidated the mechanisms of action to demonstrate the potential utility of HIDROX as a novel naturally derived virucidal agent that could contribute to stricter IAV control measures.

## 2. Materials and Methods

### 2.1. Virus

H1N1 subtype IAV strain (A/Puerto Rico/8/1934: ATCC^®^ Catalog No. VR-95) was purchased from ATCC (Manassas, VA, USA). The H3N2 subtype IAV strain (A/Hokkaido/19/98) was kindly provided by the National Institute of Animal Health (Tsukuba, Japan). The H5N1 subtype IAV strain (A/white-tailed eagle/Japan/OU-1/2022), a highly pathogenic avian influenza virus belonging to clade 2.3.4.4b, was isolated in our laboratory. Stock solutions of these viruses were diluted with phosphate-buffered saline (PBS; pH 7.4) to adjust their viral titers immediately before experiments. These diluted virus solutions served as unpurified H1, H3, and H5 IAV solutions. For protein-loading experiments, fetal bovine serum (FBS; BioWest, Bradenton, FL, USA) was added to the unpurified H1 IAV solution at a final concentration of 90%. For several experiments, the H1 IAV was purified by sucrose gradient ultracentrifugation using a Himac CS 100GXL Micro Ultracentrifuge (Hitachi High-Technologies Corporation, Tokyo, Japan) at ~140,000× *g* for 2 h with 30% and 60% sucrose solutions. The purified virus was resuspended in PBS and stored at −80 °C, which served as a purified H1 IAV solution. The purified H1 IAV solution was inactivated and dialyzed for use in an experiment measuring reactive oxygen species (ROS) and reactive nitrogen species (RNS). Briefly, the purified H1 IAV solution (viral titer: 10^7.25^ 50% tissue culture infectious dose [TCID_50_]/mL) was inactivated by reacting with 0.1% formalin (FUJIFILM Wako Pure Chemical Co., Osaka, Japan) for 48 h at 37 °C, followed by formalin neutralization with 0.6% sodium hydrosulfite (FUJIFILM Wako Pure Chemical Co.) overnight at 4 °C. Then, this solution was dialyzed in PBS at 1000 times the volume of the solution using Spectra/Por^®^ Biotech Regenerated Cellulose Dialysis Membranes MWCO: 8000 (Repligen, Boston, MA, USA) to remove the treatment chemicals. The dialysis process was repeated three times.

### 2.2. Preparation of HIDROX and HT Solutions

HIDROX^®^12% powder (1 g) was dissolved in 10 mL of PBS and centrifuged at 850× *g* for 10 min. The aqueous layer was collected and kept at −30 °C and used as the 100 mg/mL HIDROX solution. 3-Hydroxytyrosol (Tokyo Chemical Industry Co., Ltd., Tokyo, Japan) (1 g) was dissolved in 10 mL of PBS, and the 100 mg/mL HT solution was stored at −30 °C. Before experimental use, multiple concentrations of HIDROX and HT solutions were prepared by diluting the stock solutions with PBS. In some experiments, diluted HIDROX and HT were stored at 4 °C or 22 °C for 45 days before experimental use. OLIVENOL^TM^ plus+ Healing Moisturizer (Oliphenol LLC.) containing 2%, 5%, and 10% HIDROX was provided by Oliphenol LLC as the HIDROX-containing cream. The control base comprised HIDROX-free (0%) components from OLIVENOL^TM^ plus+ Healing Moisturizer. The primary components of this control base are as follows (trace components below 1% concentration are omitted): water, glycerin, stearyl alcohol, dimethicone, catearyl alcohol, glyceryl stearate, PEG-100 stearate, simmondsia chinensis (Jojoba) seed oil, sorbitan olivate, lauryl laurate, sorbitan stearate, and glycol distearate.

### 2.3. Cell Culture

Madin–Darby canine kidney (MDCK) cells were kindly provided by Dr. H. Nagano (Hokkaido Institute of Health, Sapporo, Japan). For passaging, MDCK cells were cultured in Dulbecco’s modified Eagle’s minimal essential medium (DMEM) (FUJIFILM Wako Pure Chemical Co.) supplemented with 10% FBS, 20 mM L-glutamine (FUJIFILM Wako Pure Chemical Co.), 0.15% NaHCO_3_ (FUJIFILM Wako Pure Chemical Co.), 2 μg/mL amphotericin B (Bristol-Myers Squibb Co., New York, NY, USA), and 100 μg/mL kanamycin (Meiji Seika Pharma Co., Ltd., Tokyo, Japan). After virus inoculation, cells were cultured in virus growth medium (VGM), which comprised DMEM supplemented with 0.2% bovine serum albumin (FUJIFILM Wako Pure Chemical Co.), 0.01% glucose (FUJIFILM Wako Pure Chemical Co.), 0.15% NaHCO_3_, 25 mM HEPES (FUJIFILM Wako Pure Chemical Co.), 2 μg/mL amphotericin B, and 100 μg/mL kanamycin. To incubate cells inoculated with H1 and H3 IAVs, 0.0006% trypsin (FUJIFILM Wako Pure Chemical Co.) was added to the VGM.

### 2.4. Evaluation of the Virucidal Activities of HIDROX and HT Against IAV

Unpurified H1, H3, or H5 IAV solution (viral titer: 10^7.25^ TCID_50_/mL) was mixed with HIDROX or HT solution at a 1:9 ratio. HIDROX and HT final concentrations in the mixture were 0.45, 0.90, and 4.50 mg/mL and 0.05 and 0.90 mg/mL, respectively. These concentrations were the same as those tested in our previous study, which evaluated the virucidal activities of HIDROX and HT against SARS-CoV-2 [16]. The IAV solution was also mixed with PBS, which was used as a diluent control. After incubation at 4 °C or 25 °C for 1 min to 24 h, the mixtures were added to cell cultures in VGM in a ten-fold serial dilution. After a 3-day incubation period at 37 °C, the cytopathic effect induced by IAV infection was observed, and the viral titer (TCID_50_/mL) was calculated using the Behrens–Kärber method [20]. The detection limit of the viral titer in each group is determined by the degree of cytotoxicity exhibited by the corresponding test sample. Test groups exposed to samples with higher cytotoxicity were assigned higher detection limits. The cytotoxicity of HIDROX and HT at various concentrations was assessed using the CellTiter-Glo^®^ Luminescent Cell Viability Assay (Promega, Madison, WI, USA). Based on these results, appropriate detection limits were established for each group. The percentage of viral inactivation resulting from each treatment was calculated by comparing the viral titers of the control group and test groups at the same reaction time. Specifically, if the viral titer in the control group was 10*^x^* TCID_50_/mL and that in the test group was 10*^y^* TCID_50_/mL, the inactivation rate (%) in the test group was calculated using the following formula: 100 − 100/10*^x^*^−^^*y*^.

### 2.5. Evaluation of the Virucidal Activity of HIDROX-Containing Cream Against IAV

A total of 20 mg of test cream was spread on 2.25 cm^2^ (1.5 cm × 1.5 cm) of polyethylene terephthalate film (AS ONE Co., Ltd., Osaka, Japan). The lid of a 12-well plate (Nalge Nunc International Co., Rochester, NY, USA) was inverted, and 10^7.06^ TCID_50_/60 μL of the unpurified H1 IAV solution was placed on the inverted side. The cream-coated film was used to cover the H1 IAV solution. A volume of 60 μL was sufficient to cover the entire 2.25 cm^2^ film surface without overflow. The 12-well plate was incubated from 1 min to 6 h at 25 °C, and then the viral suspension was collected. This suspension was inoculated into cells, and a ten-fold serial dilution was performed. After a 1 h incubation at 37 °C, the virus-containing cell culture medium was removed and new VGM was added. After incubation for 3 days at 37 °C, the viral titers and the percentage of viral inactivation were evaluated as described in Section 2.4.

### 2.6. Western Blotting (WB) Analysis

Purified H1 IAV solution (viral titer: 10^7.25^ TCID_50_/mL) was mixed with HIDROX or HT solution at a 1:9 ratio to final concentrations of HIDROX or HT in the mixture of 0.90 mg/mL. As a diluent control, the purified H1 IAV solution was mixed with PBS. The mixtures were placed at 25 °C for 0 (no reaction time) and 24 h. Mixtures were subsequently combined with dodecyl sulfate (SDS) buffer without or with 2-mercaptoethanol (2-Me; FUJIFILM Wako Pure Chemical Co.). Mixtures with SDS-buffer without 2-Me were used for WB to detect hemagglutinin (HA) 0; those with 2-Me were used for WB to detect HA1 and HA2 subunits of HA, NA, matrix-1 (M1), and nucleoprotein (NP). Samples were heated at 100 °C for 2 min and electrophoresed on 12% polyacrylamide gel. Precision Plus Protein™ All Blue Protein Standards (Bio-Rad Laboratories Inc., Hercules, CA, USA) were used as molecular mass markers. Electrophorized protein bands were transferred to a polyvinylidene difluoride membrane (Bio-Rad Laboratories Inc.). For WB detection of HA, NA, M1, and NP, the following primary antibodies were used: IAV H1N1 (A/Puerto Rico/8/1934) HA antibody, rabbit PAb, antigen affinity purified (Catalog No. 11684-T62, Sino Biological Inc., Beijing, China); IAV H1N1 NA antibody (Catalog No. GTX125974, Genetex Inc., Irvine, CA, USA); IAV H1N1 (A/Puerto Rico/8/34/Mount Sinai) M1 antibody, rabbit PAb, antigen affinity purified (Catalog No. 40010-T60, Sino Biological Inc.); IAV NP antibody, rabbit PAb, antigen affinity Purified (Catalog No.11675-T62, Sino Biological Inc.). Mouse anti-rabbit IgG peroxidase conjugate (Catalog No. A1949, Clone: RG-96, Sigma-Aldrich Co., Ltd., St. Louis, MO, USA) was used as a secondary antibody. Membranes treated with primary and secondary antibodies were reacted with ECL Prime Western Blotting Detection Reagent (GE Healthcare Ltd., Chicago, IL, USA) for 1 min, and the chemiluminescence was detected using the LAS-3000 Imaging System (FUJIFILM Co., Tokyo, Japan).

### 2.7. Hemagglutination Assay

Purified H1 IAV solution (viral titer: 10^7.25^ TCID_50_/mL) was mixed with HIDROX or HT solution at a 1:9 ratio to a final concentration of 0.90 mg/mL of HIDROX or HT. As a diluent control, the purified H1 IAV solution was mixed with PBS. Mixtures were incubated for 0 (no reaction time), 3, and 24 h at 25 °C. The hemagglutination assay was conducted according to the WHO Manual on Influenza Diagnosis and Surveillance [21]. Briefly, the mixtures were serially diluted two-fold in PBS in a 96-well plate, and an equal volume of 0.5% chicken erythrocytes (Japan Bio Serum Co., Tokyo, Japan) was added. Then, the plate was incubated for 30 min at 25 °C and the presence/absence of red blood cell agglutination was analyzed in each well. The hemagglutination titer (log_2_) was measured.

### 2.8. NA Assay

Purified H1 IAV solution (viral titer: 10^6.25^ TCID_50_/mL) was mixed with HIDROX or HT solution at a 1:9 ratio to a final concentration of 0.90 mg/mL of HIDROX or HT. As a diluent control, the purified H1 IAV solution was mixed with PBS. Mixtures were incubated for 24 h at 25 °C. The NA assay was conducted according to the WHO Manual on Influenza Diagnosis and Surveillance [21]. Briefly, fetuin (Sigma-Aldrich Co., Ltd.) was added to each mixture, which was then incubated for 18 h at 37 °C. The concentration of fetuin in the mixture was 6.25 mg/mL. Sialic acids are cleaved from fetuin via the NA activity of IAV. The tube containing the mixture was cooled to room temperature, the periodate reagent was added, and the tube was then incubated for 20 min. Subsequently, the arsenite reagent was added and mixed in the tube. Then, the thiobarbituric acid reagent was added, and the tube was incubated for 15 min at 100 °C. Next, the tube was placed on ice and Warrenoff reagent, composed of 95% 1-butanol and 5% concentrated hydrochloric acid, was added. The upper butanol phase was collected following centrifugation at 1600× *g* for 10 min. The optical density (O.D.) value (which reflects the amount of liberated sialic acids) of this upper phase was measured at a wavelength of 549 nm using a SmartSpec™ Plus Spectrophotometer (Bio-Rad Laboratories Inc.).

### 2.9. Real-Time Reverse Transcriptase Polymerase Chain Reaction (Real-Time RT-PCR)

Purified H1 IAV solution (viral titer: 10^7.25^ TCID_50_/mL) was mixed with HIDROX or HT in a 1:9 ratio to a final concentration of 0.90 mg/mL of HIDROX or HT. As a diluent control, the purified H1 IAV solution was mixed with PBS. Mixtures were placed at 25 °C for 0 (no reaction time), 3, and 24 h. RNA was then extracted using ISOGEN-LS (Nippon Gene, Tokyo, Japan) according to the manufacturer’s protocol. A total of 500 ng of RNA was reverse-transcribed using FastGene cDNA Synthesis 5× ReadyMix OdT (NIPPON Genetics Co., Ltd., Tokyo, Japan). Real-time PCR was performed using the Eagle-Taq Master Mix with ROX (F. Hoffmann-La Roche Ltd., Basel, Switzerland) with the following primers and probe: Avian Influenza A Matrix Forward 5′-ARATGAGTCTTCTRACCGAGGTCG-3′; Avian Influenza A Matrix Reverse 5′-TGCAAAGACATCYTCAAGYYTCTG-3′; the internal probe 6-FAM-TCAGGCCCCCTCAAAGCCGA-TAMRA [22]. Real-time PCR conditions were as follows: 1 cycle at 95 °C for 10 min, 45 cycles of 95 °C for 15 s and 60 °C for 60 s.

### 2.10. ROS and RNS Detection

The effect of HIDROX and HT on ROS and RNS levels was determined in experiments with and without the virus. Due to pathogen handling restrictions, we were unable to use viruses that had not been inactivated. In the experiment without the virus, 0.90 mg/mL HIDROX/PBS, 0.90 and 0.05 mg/mL HT/PBS, and PBS alone (diluent control) were incubated at 25 °C for 3 h. In the experiment with the virus, the inactivated/dialyzed H1 IAV solution was mixed with HIDROX or HT at a ratio of 1:9 to a final concentration of 0.90 mg/mL HIDROX or 0.90 and 0.05 mg/mL HT, and a mixture of the inactivated/dialyzed H1 IAV solution with PBS was used as the diluent control. Following incubation, the ROS and RNS levels in each solution were determined using a DCF ROS/RNS Assay Kit (biofluids, culture supernatant, cell lysates) (Abcam Ltd., Cambridge, UK). The fluorescence levels at 480 nm (excitation) and 530 nm (emission) were measured using a microplate reader (SH-9000Lab, Corona Electric Co., Ltd., Hitachinaka, Japan). The ROS/RNS concentration (μM) in each reaction was calculated based on the fluorescence of H_2_O_2_ standard samples included in the kit. When the concentration gave a negative value, this was set as 0.0 μM.

### 2.11. Evaluation of the Impact of ROS and/or RNS on HIDROX- and HT-Induced Virucidal Activity

Purified H1 IAV solution (viral titer: 10^7.25^ TCID_50_/mL) was treated with HIDROX or HT solution at a 1:9 ratio in the absence (0.00 mg/mL) or presence (0.02–0.60 mg/mL) of N-acetyl-L-cysteine (NAC; FUJIFILM Wako Pure Chemical Co.), an ROS and RNS scavenger [23]. The final concentrations of the mixtures were 0.90 mg/mL HIDROX and 0.90 and 0.05 mg/mL HT. Purified H1 IAV solution mixed with PBS was used as the diluent control. After incubation at 25 °C for 3 h, the viral titers were evaluated as described in Section 2.4. For WB, purified H1 IAV solution was treated with 0.90 mg/mL HIDROX, 0.90 mg/mL HT, or PBS in the absence or presence of 0.60 mg/mL NAC. After incubation at 25 °C for 24 h, WB targeting the IAV M1 protein was performed.

### 2.12. Transmission Electron Microscopy (TEM) Analysis of Viral Particles

The purified H1 IAV solution (viral titer: 10^5.62^ TCID_50_/mL) was mixed with HIDROX or HT solution at a 1:9 ratio, yielding final concentrations of 0.90 mg/mL HIDROX or HT. For the diluent control, purified H1 IAV solution was mixed with PBS. The mixtures were incubated at 25 °C for 24 h. The samples were applied to a 400-mesh carbon-coated collodion grid (NISSHIN EM Co., Ltd., Tokyo, Japan). The treated viruses were negatively stained for 2 min on the grids using 2% phosphotungstic acid (pH 6.5). Viral particles were then observed using a TEM (HT7700; Hitachi High-Tech Co., Tokyo, Japan).

### 2.13. Evaluation of Viral Particle Integrity Using Real-Time RT-PCR Combined with RNase Treatment

The purified H1 IAV solution (viral titer: 10^7.25^ TCID_50_/mL) was mixed with HIDROX, HT, or 70% ethanol (FUJIFILM Wako Pure Chemical Co.) at a 1:9 ratio. The final concentrations of HIDROX and HT in the mixture were 0.90 mg/mL, and the final ethanol concentration was 63%. As a diluent control, the purified H1 IAV solution was mixed with PBS. All mixtures were incubated at 25 °C for 24 h. RNase A (Roche Diagnostics, Mannheim, Germany) was then added to the mixtures at a final concentration of 2 μg/mL, followed by incubation at 37 °C for 30 min. If the integrity of the viral particles was compromised, RNase A could access and degrade the intraparticle viral RNA. Ethanol, which disrupts the envelope structure, was used as a positive control. A separate ethanol-treated group without RNase served as a negative control. Following RNase treatment, RNA was immediately extracted, and real-time RT-PCR was performed as described in Section 2.9.

### 2.14. Statistical Analysis

GraphPad Prism 8.3.4 (GraphPad Software Inc., La Jolla, CA, USA) was used for data analysis. Differences between the control and each test sample group were analyzed using Student’s *t*-test. Comparisons of the viral titer among multiple groups were performed using one-way analysis of variance (ANOVA) followed by Tukey’s multiple comparisons test or Kruskal–Wallis test followed by Dunn’s multiple comparisons test. *p* values < 0.05 were considered statistically significant.

## 3. Results

### 3.1. Virucidal Activity of HIDROX Against IAV

The unpurified H1 IAV solution was mixed with PBS or HIDROX solution. In 1 min reaction time at 25 °C, the 4.50 mg/mL HIDROX inactivated 92.50% virus (10^1.13^ TCID_50_/mL decline of viral titer when compared to the viral titer of PBS group). The 0.90 and 0.45 mg/mL HIDROX inactivated 92.50% and 84.60% virus in 30 min and 1 h reaction time, respectively (10^1.13^ and 10^0.81^ TCID_50_/mL decline) (Figure 1). Therefore, HIDROX showed concentration- and time-dependent virucidal activities against IAV.

### 3.2. Virucidal Activity of HIDROX-Containing Cream Against IAV

The 20 mg of test creams with different HIDROX contents were spread on 2.25 cm^2^ of film, and unpurified H1 IAV solution was covered with the cream-attached film. In 10 min reaction time at 25 °C, the cream containing 5% and 10% HIDROX inactivated 68.38% and 95.13% virus, respectively (10^0.50^ and 10^1.31^ TCID_50_/mL decline of viral titer when compared to the viral titer of 0% HIDROX group). In 3 h reaction time, the cream containing 2% HIDROX inactivated 97.26% virus (10^1.56^ TCID_50_/mL decline). Hence, HIDROX-containing cream showed concentration- and time-dependent IAV-inactivating activity (Figure 2).

### 3.3. Comparison of the Virucidal Activity of HIDROX and HT Against IAV

The H1 IAV-inactivating activities of 0.90 mg/mL HIDROX and 0.90 and 0.05 mg/mL pure HT solutions were compared. In the 3 h reaction time at 25 °C, 0.90 mg/mL HIDROX and 0.90 and 0.05 mg/mL HT inactivated 99.17%, 96.52%, and 93.81% virus, respectively (10^2.08^, 10^1.46^, and 10^1.21^ TCID_50_/mL decline of viral titer when compared to the viral titer of the PBS group). Statistically significant differences were observed between the HIDROX group and the 0.90 mg/mL HT group, as well as between the HIDROX group and the 0.05 mg/mL HT group. In 24 h reaction time, 0.90 mg/mL HIDROX and 0.90 mg/mL HT inactivated ≥99.99% ≥99.88 virus, respectively (≥10^3.94^ and ≥10^2.94^ TCID_50_/mL decline), and the viral titers were below the detection limit. The 0.05 mg/mL HT inactivated 99.55% virus (10^2.34^ TCID_50_/mL decline) (Figure 3). This result indicated that the virucidal activity of 0.90 mg/mL HIDROX was higher than that of 0.90 and 0.05 mg/mL HT.

The virucidal activities of HIDROX and HT solutions against H3 IAV were also evaluated. In the 3 h reaction time at 25 °C, 0.90 mg/mL HIDROX, as well as 0.90 and 0.05 mg/mL HT, inactivated 99.51%, 99.35%, and 98.67% virus, respectively (10^2.31^, 10^2.19^, and 10^1.88^ TCID_50_/mL decline). After 24 h, viral titers in all three treatment groups were below the detection limit (Appendix A). Similarly, virucidal activity was assessed against H5 IAV (a highly pathogenic avian influenza virus). Following a 3 h reaction time at 25 °C, 0.90 mg/mL HIDROX, as well as 0.90 and 0.05 mg/mL HT, inactivated 99.25%, 99.13%, and 98.46% virus, respectively (10^2.13^, 10^2.06^, and 10^1.81^ TCID_50_/mL decline). After 24 h, viral titers in all three treatment groups were also below the detection limit (Appendix A). These results indicate that HIDROX and HT exhibit comparable virucidal activity against multiple subtypes of IAV strains.

### 3.4. Virucidal Activities of HDROX and HT Against IAV Under Various Conditions

The effect of organic matter on the virucidal activities of HIDROX and HT was evaluated. Unpurified H1 IAV solutions, with or without 90% FBS, were mixed with PBS, HIDROX, or HT solutions at a 1:9 ratio, resulting in final FBS concentrations of 0% or 9%. After a 3 h reaction time at 25 °C, in the absence of FBS, 0.90 mg/mL HIDROX, as well as 0.90 and 0.05 mg/mL HT, inactivated 99.25%, 99.00%, and 97.26% virus, respectively (10^2.13^, 10^2.00^, and 10^1.56^ TCID_50_/mL decline). Moreover, in the presence of 9% FBS, 0.90 mg/mL HIDROX, as well as 0.90 and 0.05 mg/mL HT, inactivated 98.85%, 99.00%, and 95.13% virus, respectively (10^1.94^, 10^2.00^, and 10^1.31^ TCID_50_/mL decline) (Figure 4A). These results indicate that the virucidal activities of HIDROX and HT are not substantially reduced in the presence of 9% FBS.

The virucidal activities of HIDROX and HT were compared at 25 °C and 4 °C. After a 24 h reaction time at 25 °C, 0.90 mg/mL HIDROX, as well as 0.90 and 0.05 mg/mL HT, inactivated ≥99.95%, ≥99.76%, and 99.44% virus, respectively (≥10^3.31^, ≥10^2.63^, and 10^2.25^ TCID_50_/mL decline). At 4 °C, the same treatment resulted in 98.85%, 99.25%, and 96.35% inactivation (10^1.94^, 10^2.13^, and 10^1.44^ TCID_50_/mL decline) (Figure 4B). These results indicate that the virucidal activities of HIDROX and HT are reduced at lower temperatures.

The effect of long-term storage on the virucidal activities of HIDROX and HT was evaluated. HIDROX and HT were stored at either 22 °C or 4 °C for 45 days, and their virucidal activities were compared to those of freshly prepared samples (0 day, no storage). After a 3 h reaction time at 25 °C, 0.90 mg/mL HIDROX, as well as 0.90 and 0.05 mg/mL HT (0 day), inactivated 98.53%, 93.19%, and 90.00% virus, respectively, (10^1.83^, 10^1.17^, and 10^1.00^ TCID_50_/mL decline). Following 45 days of storage at 22 °C, the same formulations showed identical levels of inactivation. After 45 days of storage at 4 °C, 0.90 mg/mL HIDROX, as well as 0.90 and 0.05 mg/mL HT, inactivated 93.19%, 90.00%, and 78.46% virus, respectively, (10^1.17^, 10^1.00^, and 10^0.67^ TCID_50_/mL decline) (Figure 4C). These findings indicate that the virucidal activities of HIDROX and HT remain relatively stable even after extended storage.

### 3.5. Impact of HIDROX and HT on IAV Structural Proteins

To evaluate the impact of HIDROX and HT on IAV structural proteins (HA [HA0, HA1, and HA2 subunits], NA, M1, and NP) were analyzed with WB. The purified H1 IAV solutions were treated with 0.9 mg/mL HIDROX and 0.9 mg/mL HT for 0 and 24 h before the analysis. There were no differences in HA, NA, M1, and NP band patterns and intensities among PBS, HIDROX, and HT treatments at 0 h (Figure 5A–E). HIDROX treatment decreased the intensity of ~75 kDa band of HA0 in 24 h. In addition, HIDROX treatment increased the intensity of ~250 kDa bands. On the other hand, these band patterns and intensities in HT treatment were comparable to those in PBS treatment in 24 h (Figure 5A). In comparison to PBS treatment, HIDROX and HT treatments decreased the intensities of ~50 kDa band of HA1 subunit and ~25 kDa band of HA2 subunit in 24 h. In addition, multiple bands with >75 kDa appeared by HIDROX treatment, but not by HT treatment (Figure 5B). As a positive control, HA of H1 IAV treated with 0.18% sodium hypochlorite (NaClO) solution—shown to exhibit potent virucidal activity (Appendix A)—was also analyzed through WB. The HA bands disappeared following NaClO treatment (Appendix A). In comparison to PBS treatment, HIDROX and HT treatments induced dramatic reduction of NA band intensity at 24 h (Figure 5C). In comparison to the PBS treatment, HIDROX treatment, but not HT treatment, decreased the intensity of ~25 kDa band of M1 at 24 h. In addition, multiple bands with >50 kDa appeared by HIDROX and HT treatment. The intensity of these bands was higher in HIDROX treatment than in HT treatment (Figure 5D). In comparison to PBS treatment, HIDROX and HT treatments decreased the intensity of ~55 kDa band of NP in 24 h. In addition, ~100 kDa bands appeared by HIDROX and HT treatment. The intensity of these bands was stronger in HIDROX treatment than in HT treatment (Figure 5E).

### 3.6. Impact of HIDROX and HT on Hemagglutination and NA Activities of IAV

The impact of HIDROX and HT on the hemagglutination activity of HA, which plays a role in the binding to the virus receptor on host cells, was evaluated. In 0, 3, and 24 h reaction times, the hemagglutination titers in HIDROX and HT groups were comparable to that in the PBS group (Figure 6A). This result suggests that HIDROX and HT do not impact the hemagglutination activity of HA. The impact of HIDROX and HT on the NA activity, which plays a role in releasing the progeny viruses from infected host cells by cleaving the terminal sialic acid residues, was also evaluated. At the 24 h reaction time, the NA activities of HIDROX- and HT-treated viruses were slightly lower than that of PBS-treated viruses (Figure 6B). This result suggests that HIDROX and HT have a limited suppressive impact on NA activity.

### 3.7. Impact of HIDROX and HT on the IAV Genome

To evaluate the impact of HIDROX and HT on the IAV genome, the viral RNA was extracted from PBS-, HIDROX-, and HT-treated purified H1 IAV, and then real-time RT PCR targeting the IAV M gene was performed. In 0, 3, and 24 h reaction times, the cycle threshold (Ct) values were similar among all treatment groups (Figure 7). As a positive control, viral RNA from H1 IAV treated with 0.18% NaClO solution was also analyzed. After 24 h of reaction time, the Ct value in the NaClO-treated group was higher than that in the PBS and HIDROX groups (Appendix A).

### 3.8. Impact of ROS and/or RNS on HIDROX- and HT-Induced Virucidal Action

Polyphenols play a dual role as antioxidants and pro-oxidants [24]. We evaluated the presence or absence of ROS and/or RNS in solutions containing HIDROX and HT. In the experiments without the virus, the estimated concentrations of ROS and/or RNS were 0.0, 0.5, 17.2, and 1.5 μM in PBS, 0.90 mg/mL HIDROX/PBS, 0.90 mg/mL HT/PBS, and 0.05 mg/mL HT/PBS, respectively, after 3 h of incubation at 25 °C (Figure 8A). When the solutions were observed with the naked eye, only that of 0.90 mg/mL HT/PBS was colored pink (Appendix A). In the experiments with the virus, the estimated concentrations of ROS and/or RNS were 0.0, 0.6, 16.9, and 1.3 μM in the H1 IAV solutions mixed with PBS, 0.90 mg/mL HIDROX, 0.90 mg/mL HT, and 0.05 mg/mL HT, respectively, after 3 h of incubation (Figure 8B). Only the H1 IAV solution mixed with 0.90 mg/mL HT was colored pink (Appendix A). However, in the presence of NAC, the solution containing 0.09 mg/mL HT was transparent in the experiments both with and without the virus (Appendix A). To confirm the impact of ROS and/or RNS on virus inactivation, we evaluated the IAV-inactivating activities of 0.90 mg/mL HIDROX and 0.90 and 0.05 mg/mL HT solutions after 3 h of incubation in the absence (0.00 mg/mL) and presence (0.02–0.60 mg/mL) of NAC. The virucidal activity of HIDROX was unaffected by the presence of NAC. In contrast, NAC inhibited the virucidal activity of 0.90 and 0.05 mg/mL HT in a concentration-dependent manner (Figure 8C). To confirm the impact of ROS and/or RNS on a viral structural protein, the purified H1 IAV solution was incubated with HIDROX or HT in the absence or presence of NAC for 24 h, followed by WB targeting the IAV M1 protein. The presence of 0.60 mg/mL NAC did not affect the appearance of bands >50 kDa induced by HIDROX treatment. However, NAC treatment led to the absence of such bands induced by HT treatment (Figure 8D).

### 3.9. Effect of HIDROX and HT on Viral Particle Integrity

The viral particle protects viral RNA by enclosing it within a structural envelope. The impact of treatment with HIDROX and HT on the structural integrity of viral particles was evaluated. The morphologies of H1 IAV treated with PBS, HIDROX, or HT were observed using TEM. After a 24 h reaction time at 25 °C, multiple viral particles with clearly visible spike proteins and intact envelopes were observed in all three treatment groups. No distinct morphological differences were observed among the treatments (Figure 9A). Viral particle integrity was further examined using real-time RT-PCR following RNase treatment. Purified H1 IAV samples treated with PBS, HIDROX, HT, or ethanol (positive control) were incubated for 24 h at 25 °C before adding RNase. An ethanol-treated IAV sample without subsequent RNase treatment was also included as a negative control. Real-time RT-PCR targeting the IAV M gene revealed that the Ct values in the PBS + RNase, HIDROX + RNase, and HT + RNase groups were comparable to that of the ethanol only (no RNase) group. Conversely, the Ct value in the ethanol + RNase group was higher than that in the other four groups (Figure 9B). These findings indicate that HIDROX and HT treatments did not cause significant disruption to viral particles, unlike ethanol, which clearly compromised particle integrity.

## 4. Discussion

In this study, HIDROX and HT solutions showed concentration-dependent virucidal activities against H1 IAV (Figure 1 and Figure 3). Furthermore, the extent of viral titer reduction by HIDROX and HT treatments increased with increasing reaction time. Although a strict linear relationship between their virus inactivation efficiencies and reaction time was not observed, HIDROX and HT solutions clearly showed potent virucidal activities in longer reaction time (Figure 1 and Figure 3). These results indicate that the IAV-inactivating activities of HIDROX and HT are time-dependent. Previously, we demonstrated a similar level of virucidal activity of HIDROX against SARS-CoV-2 that required reaction times of 30 min to 24 h [16]. Thus, HIDROX may exhibit virucidal activity against multiple enveloped viruses within relatively short reaction times. The HIDROX-containing cream also showed concentration- and time-dependent virucidal efficacy against H1 IAV, and the 10% HIDROX-containing cream inactivated 95.13% IAV in 10 min (Figure 2). This 10% HIDROX-containing cream also inactivated 94.37% SASR-CoV-2 in 10 min [16]. Although our study showed that both the HIDROX-containing solution and cream exhibited IAV-inactivation activity, the viral titers of the H1 IAV solutions used in Figure 1 and Figure 2 differed. Consequently, a direct comparison of the virucidal activities of the HIDROX-containing solution and cream could not be performed. In a previous evaluation of the safety of HIDROX, an in vivo toxicity test by oral administration of HIDROX to rats for 90 days showed that a dosage of 2000 mg/kg/day did not induce signs of toxicity [19]. Furthermore, an alginate bilayer film containing HIDROX exhibited no skin irritation [17]. Although the concentrations of HIDROX and HT that cause skin toxicity were not tested in this study, they should be evaluated in the future using in vivo studies, such as skin irritation tests, because the possible high biocompatibility of HIDROX may be advantageous to use as virucidal agent and hand cream.

IAV transmission occurs primarily through short-range exposure to droplets and aerosols, with no conclusive evidence supporting transmission through contaminated hands [25]. However, studies have isolated infectious IAV from human fingers [25,26], and hand hygiene has been associated with reduced influenza incidence [25]. While current evidence suggests hand-mediated IAV transmission is likely limited, this route cannot be entirely excluded. Therefore, influenza control guidelines recommend hand hygiene [27]. HIDROX-containing cream may help mitigate the potential IAV transmission risk through hand contact.

Olive oil contains three main polyphenols: oleuropein, HT, and tyrosol. Notably, HT is the major phenolic component in olive oil and is reported to have antioxidant, anti-inflammatory, antimicrobial, anticarcinogenic, neuroprotective, and anti-atherosclerosis activities [14,15]. HIDROX is reported to show higher biological potency than that of pure HT in several studies. For instance, HIDROX exhibited greater neuroprotective efficacy than HT did in the *Caenorhabditis elegans* model of Parkinson’s disease [28]. In addition, HIDROX showed protective activity against cuprizone-induced brain degeneration in the zebra fish model, whereas HT did not show such a protective efficacy [29]. Furthermore, olive vegetation water, but not HT, inhibited lipopolysaccharide-induced tumor necrosis factor-α production from a human monocytic cell line [30]. Our previous study showed that HIDROX exhibited more potent SARS-CoV-2-inactivating activity than that of pure HT [16]. Consistent with these reports, HIDROX also showed stronger H1 IAV-inactivating activity than that of pure HT in the present study (Figure 3). Furthermore, HIDROX and HT showed virucidal activity against H3 and H5 IAV strains (Appendix A). Our previous study similarly showed that HT exhibited virucidal activity against multiple IAV strains with different HA subtypes [12]. This broad-spectrum activity of HIDROX and HT may be advantageous for IAV control, given the concurrent circulation of multiple subtypes of IAV strains in nature. In the previous studies, a tea-leaf-derived extract containing multiple polyphenols showed stronger IAV-inactivation activity than that of the individual polyphenols contained in the extract [10]. Thus, the additive action of multiple virucidal compounds, including HT, may have contributed to the comprehensive IAV- and SARS-CoV-2-inactivating activities of HIDROX.

Environmental factors, such as organic matter and temperature, as well as prolonged storage, affect the efficacy of virucidal agents [31]. In this study, HIDROX and HT maintained their virucidal activity even in the presence of organic matter (Figure 4A). This distinguishes them from many existing disinfectants whose virus inactivation activities are inhibited by organic matter. However, their virucidal activity decreased at low temperature (Figure 4B), indicating that application temperature should be considered, as with other temperature-sensitive virucidal agents. Both HIDROX and HT retained virucidal activities after 45 days of storage at 22 °C and 4 °C (Figure 4C), indicating favorable stability for practical use. These findings provide valuable information for optimizing the application of HIDROX and HT.

The WB analysis showed that HIDROX and HT exposure for 24 h induced the appearance of high molecular mass bands and the disappearance of target bands, which suggested the occurrence of structural abnormalities or destruction of spike proteins, HA and NA, and internal proteins, M1 and NP (Figure 5). However, the information obtainable from WB analysis alone is limited, and the nature of these high-molecular-mass bands requires further exploration. The impact of HIDROX on these viral proteins was stronger than that of HT, which may have been due to the additive activity of HT and other compounds present in HIDROX. HIDROX and HT also induced the appearance of high-molecular-mass bands in WB that assessed the expression of the spike protein of SARS-CoV-2 [16]. Another polyphenol-enriched plant extract has been shown to increase the intensities of high molecular mass bands in WB assessing the expression of HA of IAV [10]. Polyphenols were reported to interact with proteins noncovalently and covalently, which may induce the aggregation of the polyphenol–protein complexes with high molecular mass [32]. An in silico docking simulation study showed that oleuropein and HT in olive leaf extract could bind to the hydrophobic pocket of envelope protein gp41 of HIV-1 [33,34]. This binding of oleuropein and HT blocked the formation of a six helical bundle fusion complex in gp41, which inhibited HIV-1 infection [34]. Thus, the binding of phenolic compounds to proteins can inhibit their actions, although HIDROX and HT did not impact on the hemagglutination activity of IAV (Figure 6A). This finding was consistent with our previous studies showing that HT did not affect this activity of IAV [12]. These results suggest that the action sites of HIDROX and HT on HA are not involved in hemagglutination activity. The HA1 and HA2 subunits of HA are involved in the binding of virions to sialic acids on host cells and the fusion of virus envelope and endosomal membrane of host cells, respectively [35]. The reduced intensity of the HA2 band following treatment with HIDTOX and HT might affect membrane fusion function. However, WB results alone cannot confirm this hypothesis, and additional functional assays of HA2 are required. This represents a limitation of our study. Although HIDROX and HT treatment induced the disappearance of NA band in WB (Figure 5C), the inhibitory effect of HIDROX and HT to NA activity were limited (Figure 6B). The result of NA assay was generally consistent with our previous studies showing that HT did not affect to NA activity of IAV [12]. The results of the WB and NA assay suggest that HIDROX and HT destroy or block the epitope regions targeted by anti-NA antibodies used in the WB, but the region involved in NA activity remained largely intact. Furthermore, the results of WB targeting M1 and NP suggest that HIDROX and HT impacted both the external spike proteins and the internal proteins of IAV (Figure 5D,E).

No changes of Ct values were induced by HIDROX and HT treatments in real-time RT-PCR analysis targeting the IAV M gene (Figure 7). In contrast, reduction of Ct values by HIDROX and HT treatments were observed in real-time RT-PCR analysis targeting the SARS-CoV-2 N gene [16]. These results suggest that the extent of the impact of HIDROX and HT on the virus genome differs among different virus species. The possible factors influencing such a discrepancy may be the differences in the nucleotide sequences of viral genomes and in the structure and robustness of virions that can affect the accessibility of HIDROX and HT to the virus genome.

Although the antioxidant function of polyphenols, including HT, is well known, they have also been reported to exhibit pro-oxidative effects in some circumstances [24]. In this study, HT in PBS generated ROS and/or RNS (Figure 8A,B). Although the possibility that N_2_ was supplied from the air cannot be completely excluded, we speculate that ROS, rather than RNS, was produced in this experimental system, in which a nitrogen source was thought to be minimal. Additionally, the color of the solutions containing HT changed to pink (Appendix A), implying the partial oxidation of HT. HT oxidizes into HT-quinones, which generate superoxide through the semiquinone–quinone cycle [36]. This oxidation process occurs not only in the presence of oxidases or oxidizing agents but also in the presence of PBS alone [37]. Such an oxidation reaction may have occurred under our experimental conditions, and the generated superoxide may have contributed to the HT-induced virus inactivation. The addition of NAC prevented the color change of the solutions containing HT (Appendix A) and reduced the virucidal activity of HT (Figure 8C). NAC reacts with HT-quinone through Michael addition, forming the NAC adduct of HT [23], which is also an antioxidant. Thus, NAC addition may have suppressed the semiquinone–quinone cycle initiated by HT-quinone and reduced superoxide generation. Unfortunately, we were unable to clarify whether or not ROS and/or RNS induced by HT was scavenged by NAC with the ROS/RNS assay kit used in this study due to an unknown measurement issue. Another analysis should be performed to clarify the role of NAC in ROS and/or RNS scavenging. WB results showed that NAC prevented the formation of abnormal viral protein bands with high molecular mass induced by HT (Figure 8D). A quinone was reported to bind proteins and change their functions [38]. In our experiment, NAC may have prevented the protein aggregation caused by HT-quinone and/or superoxide. In contrast to the results with 0.9 mg/mL HT, 0.9 mg/mL HIDROX only induced the production of slight amounts of ROS and/or RNS, and NAC did not inhibit its virucidal mechanism. The ROS and/or RNS concentration in the 0.9 mg/mL HIDROX group was lower than that in the 0.05 mg/mL HT group, suggesting that quinone formation and superoxide production were suppressed by antioxidant compounds contained in HIDROX. Therefore, the virucidal activity of HIDROX may be mediated by a mechanism different from that of HT alone.

Previous TEM observations of HT-treated H9N2 subtype IAV showed that 0.20 mg/mL HT-induced viral particle disruption occurred after a 24 h reaction time [12]. However, in this study, TEM analysis did not provide clear evidence of H1 IAV particle destruction by 0.9 mg/mL HIDROX or HT treatments (Figure 9A). This discrepancy may result from differences in IAV subtypes or experimental conditions. Real-time RT-PCR with RNase treatment also failed to detect H1 IAV particle integrity disruption by HIDROX and HT treatments (Figure 9B). WB analyses revealed abnormal band patterns in surface spike and internal viral proteins. The HIDROX- and HT-induced spike protein abnormalities detected through WB analyses may have caused infectivity loss, despite being undetectable by TEM. Alternatively, unidentified compounds in HIDROX, HT itself, or HT-induced superoxide might create undetectable envelope pores (too small for TEM or real-time RT-PCR/RNase detection), or penetrate through the envelope, which allow entry into viral particles without visible envelope damage, potentially damaging internal structural proteins. While this study shows that HIDROX and HT possess IAV-inactivation activity and partially elucidates its mechanism, further analysis is required to reconcile all mechanism findings without contradictions.

## 5. Conclusions

In the present study, HIDROX was found to have potent virucidal activity in a concentration- and time-dependent manner in a solution and cream forms against IAV. In addition, the virucidal activity was more potent than that of pure HT. One of the possible mechanisms of virucidal activities of HIDROX may involve the induction of structural abnormalities or destruction of the structural proteins of IAV. The virucidal action of pure HT may depend on its oxidation and production of superoxide, but that of HIDROX does not. The current study showed the possible utility of HIDROX as a naturally derived safe IAV virucidal agent. The present findings may have exhibited the potential applicability of HIDROX in the forms of disinfectant, body/hand cream, mouth wash, and lozenges. Evaluation of the therapeutic efficacy of HIDROX administration (e.g., local respiratory delivery) in vivo is a matter for future study. Here, we showed the potential application of HIDROX as a naturally derived virucidal agent that could help improve current IAV infection control measures.

## 6. Patents

H.O. and Y.T. are the inventors of PCT/US2021/062213: Antiviral Olive Extract Compositions and Methods.

## Figures and Tables

**Figure 1 pathogens-14-00529-f001:**
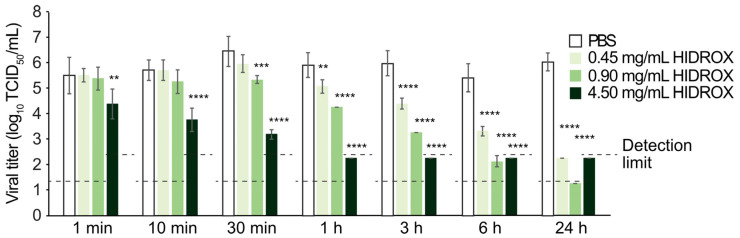
Virucidal activity of HIDROX against IAV. Unpurified H1 IAV solution was mixed with PBS or HIDROX solution, and the mixtures were incubated at 25 °C for 1 min to 24 h. The viral titer in each group is shown. The detection limits (dashed lines) of the viral titer were 10^1.25^ TCID_50_/mL in the PBS, 0.45 and 0.90 mg/mL in the HIDROX groups, and 10^2.25^ TCID_50_/mL in the 4.50 mg/mL HIDROX group. Results are shown as mean ± SD (*n* = 7–8 per group). Student’s *t*-test was performed to evaluate the statistically significant difference between the PBS group and each HIDROX group; ** *p* < 0.01; *** *p* < 0.001; **** *p* < 0.0001.

**Figure 2 pathogens-14-00529-f002:**
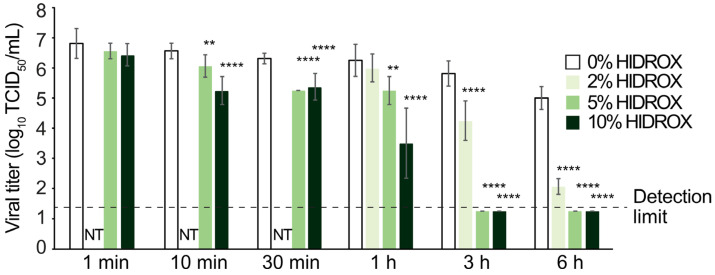
Virucidal activity of HIDROX-containing cream against IAV. Unpurified H1 IAV solution was covered with 0%, 2%, 5%, and 10% HIDROX-containing-cream-attached films and incubated for 1 min to 6 h at 25 °C. After the indicated reaction times, the viral solutions were recovered, and the viral titers were measured. The detection limit (dashed lines) of the viral titer was 10^1.25^ TCID_50_/mL in all groups. Results are indicated as mean ± SD (*n* = 8 per group). Student’s *t*-test was performed to evaluate the statistically significant difference between the 0% HIDROX group and each test group; ** *p* < 0.01; **** *p* < 0.0001; NT: not tested.

**Figure 3 pathogens-14-00529-f003:**
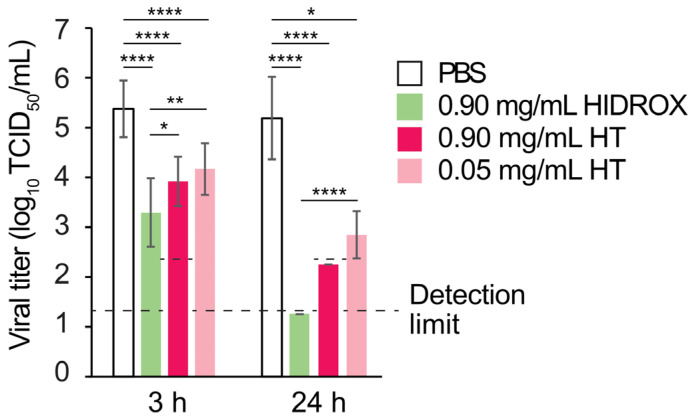
Comparison of the virucidal activity of HIDROX and HT against IAV. Unpurified H1 IAV solution was mixed with PBS, HIDROX, or HT solution. Mixtures were incubated at 25 °C for 3 and 24 h. The titer in each group is shown. The detection limits (dashed line) of the viral titer were 10^1.25^ TCID_50_/mL in the PBS, 0.90 mg/mL HIDROX, and 0.05 mg/mL HT groups, and 10^2.25^ TCID_50_/mL in the 0.90 mg/mL HT group. Results are indicated as mean ± SD (*n* = 12–16 per group). One-way ANOVA followed by Tukey’s multiple comparisons test (for 3 h), and Kruskal–Wallis test followed by Dunn’s multiple comparisons test (for 24 h), were performed; * *p* < 0.05, ** *p* < 0.01; **** *p* < 0.0001.

**Figure 4 pathogens-14-00529-f004:**
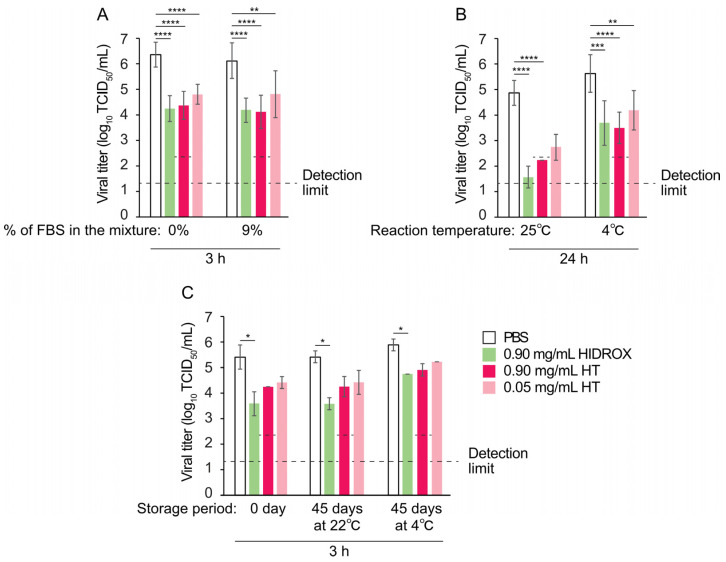
Virucidal activities of HIDROX and HT under various conditions. (**A**) Unpurified H1 IAV solution without or with FBS was mixed with PBS, HIDROX, or HT solution. Mixtures were incubated at 25 °C for 3 h. Results are indicated as mean ± SD (*n* = 8 per group). One-way ANOVA followed by Tukey’s multiple comparisons test was performed. (**B**) Unpurified H1 IAV solution was mixed with PBS, HIDROX, or HT solution. Mixtures were incubated at 25 °C or 4 °C for 24 h. Results are indicated as mean ± SD (*n* = 8 per group). Kruskal–Wallis test followed by Dunn’s multiple comparisons test (for the 25 °C condition), and one-way ANOVA followed by Tukey’s multiple comparisons test (for the 4 °C condition), were performed. (**C**) Unpurified H1 IAV solution was mixed with PBS, HIDROX, or HT solution (0 day, 45 days at 22 °C, and 45 days at 4 °C). Mixtures were incubated at 25 °C for 3 h. Results are indicated as mean ± SD (*n* = 3 per group). Kruskal–Wallis test followed by Dunn’s multiple comparisons test was performed; ** p* < 0.05; ** *p* < 0.01; *** *p* < 0.001; **** *p* < 0.0001.

**Figure 5 pathogens-14-00529-f005:**
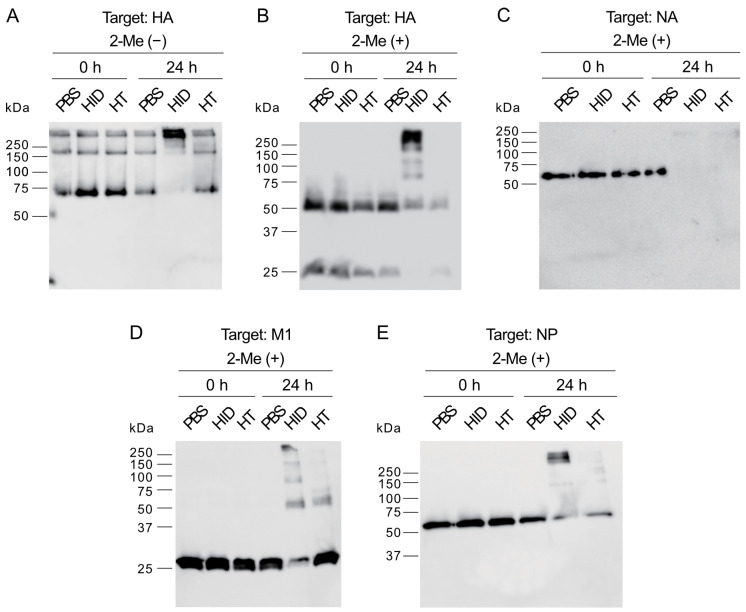
Impact of HIDROX and HT on IAV structural proteins. (**A**–**E**) Purified H1 IAV solution was mixed with PBS, HIDROX, or HT solution. Mixtures were incubated at 25 °C for 0 and 24 h. Images are the results of WB for HA0 (**A**), HA1 and HA2 (**B**), NA (**C**), M1 (**D**), and NP (**E**). HID: HIDROX.

**Figure 6 pathogens-14-00529-f006:**
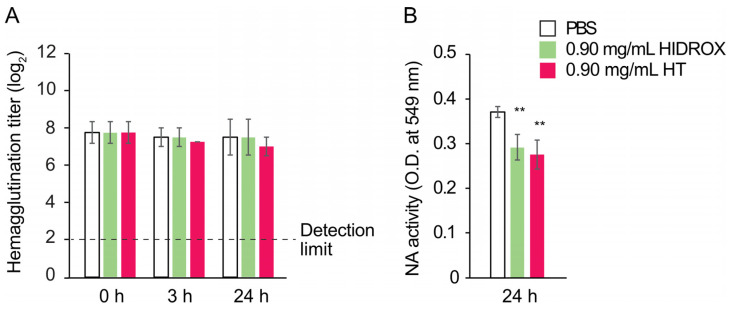
Impact of HIDROX and HT on hemagglutination and NA activities of IAV. (**A**,**B**) Purified H1 IAV solution was mixed with PBS, HIDROX, or HT solution. (**A**) Mixtures were incubated at 25 °C for 0, 3, and 24 h, and the hemagglutination titer was then evaluated. The detection limit of the hemagglutination titer was 2 log_2_ and represented as a dashed line. (**B**) Mixtures were incubated at 25 °C for 24 h. The NA activity (O.D. at 549 nm) of the mixture was measured. (**A**,**B**) Results are indicated as mean ± SD (*n* = 4 per group). Student’s *t*-test was performed to evaluate the statistically significant difference between the PBS group and each test solution group; ** *p* < 0.01.

**Figure 7 pathogens-14-00529-f007:**
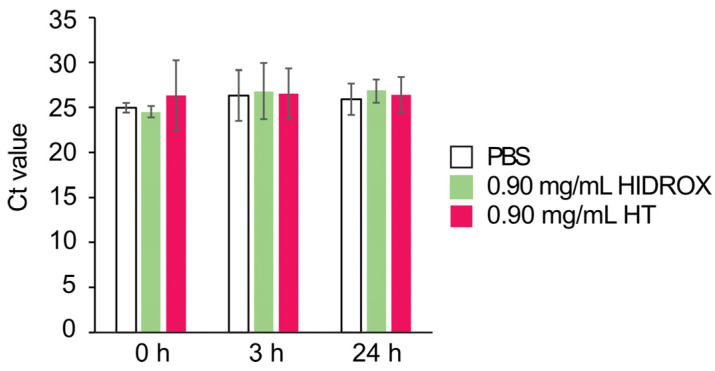
Impact of HIDROX and HT on the IAV genome. Purified H1 IAV solution was mixed with PBS, HIDROX, or HT solution. Mixtures were incubated at 25 °C for 0, 3, and 24 h. Real-time RT-PCR targeting the IAV M gene was performed, and the Ct value was evaluated. Results are indicated as mean ± SD (*n* = 8 per group). Student’s *t*-test was performed to evaluate the statistically significant difference between the PBS group and each test solution group.

**Figure 8 pathogens-14-00529-f008:**
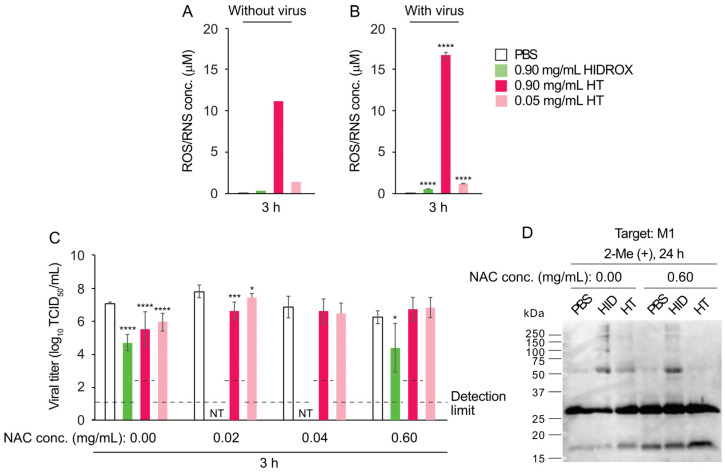
Impact of ROS and/or RNS on HIDROX- and HT-induced virucidal action. (**A**) PBS, HIDROX/PBS, and HT/PBS were incubated at 25 °C for 3 h (*n* = 1 per group). (**B**) The inactivated/dialyzed H1 IAV solution was mixed with PBS, HIDROX, or HT solution. Results are indicated as mean ± SD (*n* = 3 per group). (**A**,**B**) Estimated concentration of ROS and RNS in each solution was measured. (**C**) Purified H1 IAV solution was mixed with PBS, HIDROX, or HT solution. Mixtures were incubated at 25 °C for 3 h in the absence or presence of NAC. The viral titer is shown for each group. The detection limits (dashed line) of the viral titer were 10^1.25^ TCID_50_/mL in the control (PBS), 0.90 mg/mL HIDROX, and 0.05 mg/mL HT groups and 10^2.25^ TCID_50_/mL in the 0.90 mg/mL HT group. (**B**,**C**) Student *t*-test was performed to evaluate the statistically significant difference between the PBS group and each test solution group; * *p* < 0.05, *** *p* < 0.001, **** *p* < 0.0001. NT: not tested. (**D**) Purified H1 IAV solution mixed with PBS, 0.90 mg/mL HIDROX, or 0.90 mg/mL HT solution was incubated at 25 °C for 24 h in the absence or presence of NAC. The image shows the WB results for M1. HID: HIDROX.

**Figure 9 pathogens-14-00529-f009:**
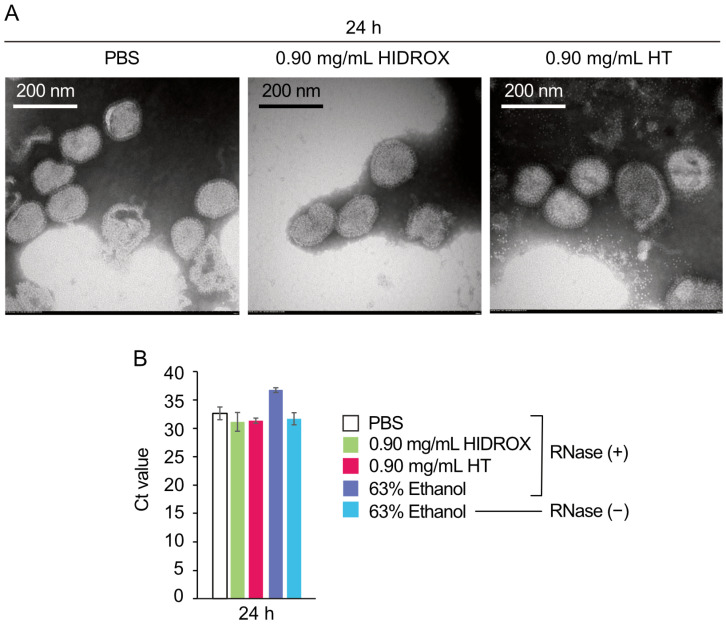
Effect of HIDROX and HT on viral particle integrity. (**A**) The purified H1 IAV solution was mixed with PBS, HIDROX, or HT solution. The mixtures were incubated at 25 °C for 24 h. The viral particles were observed using TEM. (**B**) Purified H1 IAV solution was mixed with PBS, HIDROX, HT, or ethanol and incubated for 24 h before treatment with RNase (negative control: ethanol without RNase treatment). Real-time RT-PCR targeting the IAV M gene was performed, and the Ct value was evaluated. Results are indicated as mean ± SD (*n* = 2 per group).

## Data Availability

All data supporting the findings of this study are included in the manuscript. The data analyzed during the present study are available from the corresponding author upon reasonable request.

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
