# Peer review of "In Vitro Influenza A Virus-Inactivating Activity of HIDROX®, Hydroxytyrosol-Rich Aqueous Olive Pulp Extract"

_pathogens, 2025, doi:10.3390/pathogens14060529_

Round 1
Reviewer 1 Report
Comments and Suggestions for Authors
This paper investigates the in vitro inactivation activity of hydroxytyrosol-rich olive fruit pulp water extract (HIDROX®) against influenza A virus (IAV) and its mechanism of action. The study found that HIDROX® exhibited concentration- and time-dependent inactivation activity, which was superior to that of pure hydroxytyrosol (HT). After treatment with HIDROX® and HT, the band intensity of viral structural proteins weakened, and new bands appeared at higher molecular weights, indicating that structural changes or abnormalities in viral proteins may have been induced. The effects of HIDROX® and HT on hemagglutinin and neuraminidase activity were limited, and they had no significant impact on the viral genome. HT treatment produced high levels of reactive oxygen species (ROS) and/or reactive nitrogen species (RNS), while HIDROX® treatment did not exhibit this phenomenon. The results suggest that HIDROX®, as a naturally derived IAV inactivator, has potential application value. However, several issues need to be addressed before publication:
1. Were the sample size and number of repetitions sufficient in the experiment to ensure statistical significance and reproducibility of the results? Especially in experiments with different concentrations and time points, were there enough repeated experiments to support the conclusions?
2. In addition to using PBS as a control, was consideration given to using other compounds known to have inactivation activity as positive controls to better assess the effects of HIDROX® and HT?
3. Were the high molecular weight bands observed in the Western blot analysis further analyzed to determine their nature? For example, were mass spectrometry analyses or other methods conducted to confirm whether these bands are aggregates or degradation products of viral proteins?
4. Mechanism of action of ROS and RNS: How do the ROS and RNS produced by HT treatment specifically affect the structure and function of viral proteins? Can other experimental methods (such as electron microscopy observation) be used to further verify the direct effects of these reactive substances on viral particles?
5. Stability of HIDROX® and HT under different storage conditions: How stable are HIDROX® and HT under different storage conditions? Does their inactivation activity change after long-term storage? This is very important for their practical application as disinfectants.
6. Testing against other influenza virus subtypes: Has the study considered conducting similar inactivation experiments on other influenza virus subtypes (such as H3N2, H5N1, etc.) to assess the broad-spectrum activity of HIDROX® and HT?
7. Although the paper mentions that HIDROX® showed low toxicity in in vitro and in vivo experiments, has a detailed safety assessment of its skin irritation and allergic reactions been conducted for practical applications, especially when used as hand cream or disinfectant?
8. In practical applications, will the inactivation activity of HIDROX® and HT be affected by environmental factors (such as temperature, humidity, presence of organic matter, etc.)? Have relevant experiments been conducted to assess the impact of these factors on their effectiveness?

Author Response
We are grateful to the reviewers for their valuable comments and helpful suggestions. We have revised the manuscript based on the reviewers’ comments. All the changes in the revised manuscript are highlighted in red color font. Further, we have provided our point-by-point responses to the reviewer’s comments below:
Reviewer #1:
Comment 1. Were the sample size and number of repetitions sufficient in the experiment to ensure statistical significance and reproducibility of the results? Especially in experiments with different concentrations and time points, were there enough repeated experiments to support the conclusions?
Response 1. We believe that sufficient samples were tested in each experiment to conduct statistical analyses. Each experiment was replicated at least once. The sample size for each figure—representing the total number of samples analyzed across all replicates—was provided in the figure legends.
Comment 2. In addition to using PBS as a control, was consideration given to using other compounds known to have inactivation activity as positive controls to better assess the effects of HIDROX® and HT?
Response 2. In response to the reviewer’s comment, we conducted an additional experiment comparing the effects of HIDROX and sodium hypochlorite (NaCIO) solution, a known virucidal disinfectant that damages the influenza A virus (IAV) proteins and viral genome. These results are presented in Supplementary Figure 2. Our findings indicate that the IAV inactivation mechanisms of HIDROX and NaClO may differ. Additionally, we included ethanol as a positive control, disrupting the viral envelope structure (Figure 9B). This result further indicates that the IAV inactivation mechanisms of HIDROX and ethanol are also distinct.
Comment 3. Were the high molecular weight bands observed in the Western blot analysis further analyzed to determine their nature? For example, were mass spectrometry analyses or other methods conducted to confirm whether these bands are aggregates or degradation products of viral proteins?
Response 3. We appreciate the reviewer’s valuable feedback. The analyses the reviewer suggested are insightful; however, due to time constraints during the revision period, we could not conduct them. We acknowledge this as a limitation of our study and have addressed it in lines 626–628.
Comment 4. Mechanism of action of ROS and RNS: How do the ROS and RNS produced by HT treatment specifically affect the structure and function of viral proteins? Can other experimental methods (such as electron microscopy observation) be used to further verify the direct effects of these reactive substances on viral particles?
Response 4. We thank the reviewer for their insightful comment. In response, we conducted additional experiments using transmission electron microscopy (TEM) (Figure 9A) and real-time RT-PCR with RNase treatment (Figure 9B) to assess morphological changes in IAV particles following treatment with HIDROX and hydroxytyrosol (HT). We have included relevant methodological details in Sections 2.12. and 2.13. and discussed the findings in Section 3.9. and lines 697–712 of the revised manuscript.
Comment 5. Stability of HIDROX® and HT under different storage conditions: How stable are HIDROX® and HT under different storage conditions? Does their inactivation activity change after long-term storage? This is very important for their practical application as disinfectants.
Response 5. In response to the reviewer’s comment, we conducted additional experiments to assess the stability of the virucidal activity of HIDROX and HT after 45 days of storage at 22°C and 4°C (Figure 4C). Descriptions of these experiments have been added in lines 130–131, 406–416, and 619–622.
Comment 6. Testing against other influenza virus subtypes: Has the study considered conducting similar inactivation experiments on other influenza virus subtypes (such as H3N2, H5N1, etc.) to assess the broad-spectrum activity of HIDROX® and HT?
Response 6. In response to the reviewer’s comment, we conducted additional experiments to assess the virucidal activities of HIDROX and HT against H3N2 IAV and highly pathogenic avian influenza H5N1 virus (Supplementary Figure 1A, B). Descriptions of these experiments have been added in lines 101–104, 377–387, and 603–608.
Comment 7. Although the paper mentions that HIDROX® showed low toxicity in in vitro and in vivo experiments, has a detailed safety assessment of its skin irritation and allergic reactions been conducted for practical applications, especially when used as hand cream or disinfectant?
Response 7. The 12% HIDROX-containing cream, “OLIVENOL® plus+ Skin Solutions: Healing Moisturizer,” is commercially available (source: https://www.holisticsolutionshk.com/wp-content/uploads/2020/01/From-Organic-Olive-Juice-to-OLIVENOL-plus-Pamphlet-Final-191119-with-apple-green-background.pdf). However, we could not obtain additional safety test data—including skin irritation and allergic reaction testing—for HIDROX-containing products because we could not contact the personnel in charge at Oliphenol LLC during the manuscript revision period. Consequently, we cannot provide further details under these circumstances.
Comment 8. In practical applications, will the inactivation activity of HIDROX® and HT be affected by environmental factors (such as temperature, humidity, presence of organic matter, etc.)? Have relevant experiments been conducted to assess the impact of these factors on their effectiveness?
Response 8. In response to the reviewer’s comment, we conducted additional experiments to explore the effects of environmental factors—specifically, organic matter (Figure 4A) and temperature (Figure 4B)—on the virucidal activity of HIDROX and HT. Descriptions of these experiments have been added in lines 107–109, 161, 390–405, and 613–619. The effects of humidity on virucidal activity could not be evaluated due to pathogen handling constraints. Specifically, as the virus and test solution mixture was incubated in sealed tubes, we could not control the internal humidity.

Reviewer 2 Report
Comments and Suggestions for Authors
Thank you very much for the opportunity to review the original research article "In vitro influenza A virus-inactivating activity of HIDROX®, hydroxytyrosol-rich aqueous olive pulp extract"
Here are my comments and questions:
1. The authors tested an olive aqueous extract and a product, containing the extract in various concentrations in a form of a cream. Influenza virus is transmitted via aerosol and respiratory secretions mainly. How does the application of a cream would contribute to limiting the infection in seasonal outbreaks? It would be much more useful to test forms for local respiratory administration - via nasal or inhalation route.
2. Although the authors used the same research design as in their previous publication I do not find any practical sense of 24 hours contact time of the sample with the virus. Being an enveloped, IAV is relatively unstable virus in the environment and gets inactivated itself with the time even at room temperature. Since the extract exerts its activity to a great extent at 3rd hour that would be sufficient as time contact.
3. The reduction of viral titers is expressed in percent but it is not explained how these differences were achieved. Calculation manner should be provided in Materials and Methods section.
4. What was the viral dose used in virucidal panels? The titer is indicated but did the authors use undiluted virus to mix with the extract in 1:9 ratio or the stock had already been diluted? That could be clarified.
5. There is confusion with figures 2 and 3 - texts and legends to the figures should be replaced.
6. If the authors state (in conclusions) that the extract has a therapeutic potential for treatment of influenza the antiviral action against the replication should be verified by a panel of experiments both in prophylactic and therapeutic course in cell cultures -line or primary cells.
7. Some frequent repetitions could be avoided such as in line 47, 48 and 56 - anti-influenza virus drugs by synonyms.
Author Response
We are grateful to the reviewers for their valuable comments and helpful suggestions. We have revised the manuscript based on the reviewers’ comments. All the changes in the revised manuscript are highlighted in red color font. Further, we have provided our point-by-point responses to the reviewer’s comments below:
Reviewer #2:
Comment 1. The authors tested an olive aqueous extract and a product, containing the extract in various concentrations in a form of a cream. Influenza virus is transmitted via aerosol and respiratory secretions mainly. How does the application of a cream would contribute to limiting the infection in seasonal outbreaks? It would be much more useful to test forms for local respiratory administration - via nasal or inhalation route.
Response 1. We thank the reviewer for their insightful comment. IAV is primarily transmitted through short-range exposure to droplets and aerosols, with no conclusive evidence supporting transmission through contaminated hands (doi: 10.1111/irv.12080). However, studies have isolated infectious IAV from human fingers (doi: 10.1111/irv.12080; doi: 10.1111/1469-0691.12324), and hand hygiene has been associated with reduced influenza incidence (doi: 10.1111/irv.12080). While current evidence suggests hand-mediated IAV transmission is likely limited, this route cannot be entirely excluded. Consequently, hand hygiene remains a recommended measure in influenza control guidelines (https://www.cdc.gov/flu/hcp/infection-control/healthcare-settings.html). HIDROX-containing cream may help mitigate the potential risk of IAV transmission through hands. As the reviewer suggested, we plan to evaluate the virucidal effects of local respiratory administration of HIDROX in a future in vivo study. These additions have been incorporated in lines 582–589 and 724–725.
Comment 2. Although the authors used the same research design as in their previous publication. I do not find any practical sense of 24 hours contact time of the sample with the virus. Being an enveloped, IAV is relatively unstable virus in the environment and gets inactivated itself with the time even at room temperature. Since the extract exerts its activity to a great extent at 3rd hour that would be sufficient as time contact.
Response 2. We agree with the reviewer’s comment. However, we considered the potential limitation that the assays used to analyze the virucidal mechanism of action might not detect minor abnormalities within a short reaction time. Therefore, we extended the incubation time to 24 h. We hypothesized that this experimental condition would provide a more comprehensive understanding of the mechanism by which HIDROX and HT interact with IAV.
Comment 3. The reduction of viral titers is expressed in percent but it is not explained how these differences were achieved. Calculation manner should be provided in Materials and Methods section.
Response 3. We apologize for the confusion in the original description. We have now provided a more detailed explanation in lines 170–175 and 186–188.
Comment 4. What was the viral dose used in virucidal panels? The titer is indicated but did the authors use undiluted virus to mix with the extract in 1:9 ratio or the stock had already been diluted? That could be clarified.
Response 4. Viral titers were adjusted by diluting the stock virus solutions with PBS immediately before experimental use. This detail has been added to the manuscript in lines 105–106.
Comment 5. There is confusion with figures 2 and 3 - texts and legends to the figures should be replaced.
Response 5. We apologize for this error. The corrections have been made in the revised manuscript.
Comment 6. If the authors state (in conclusions) that the extract has a therapeutic potential for treatment of influenza the antiviral action against the replication should be verified by a panel of experiments both in prophylactic and therapeutic course in cell cultures -line or primary cells.
Response 6. We agree with the reviewer’s comment. As we did not evaluate the antiviral effects of HIDROX and HT on viral replication in infected cells, we have removed the related discussion from the revised manuscript.
Comment 7. Some frequent repetitions could be avoided such as in line 47, 48 and 56 - anti-influenza virus drugs by synonyms.
Response 7. In response to the reviewer’s comment, we have revised the text in lines 49–51.

Reviewer 3 Report
Comments and Suggestions for Authors
Mohamed et al evaluated the inactivation activity of olive oil derived polyphenol against influenza A virus in vitro setting. The efficacy of HIDROX, one of the polyphenols evaluated in the present study seemed to be high. However, the authors should summarize the data more systemically to lead the authors more reasonable conclusions. Otherwise, so far in the current version of the manuscript, it must be hard to be published.
1. Throughout the manuscript, I cannot get the idea what is possible mechanism of inactivation activity of HIDROX against IAV. HIDROX will alter the construction of the viral protein, but not limited to the outer membrane proteins but internal proteins. Potential change of outer membrane protein construction did not fully affect the hemagglutination and neuraminidase activities. How about the affect to polymerase activity? Again, the authors put the “potential” inactivation mechanisms of HIDROX against IAV, but those are rarely covered by the phenomenon obtained in the present study. In addition, the authors argued the HA2 could be the target of HIDROX without indicating the apparent evidences/bases of its hypothesis. This should be also critical to obtain the conclusion of the present study.
2. The authors may conclude that the antiviral activity of HIDROX was time-dependent manner. However, according to the results in Figure2. It might not be time-dependent manner. 1.46 log10 TCID50/ml decline was observed in 3h reaction of 0.05mg/ml of HT while only 2.34 log10 TCID50/ml decline was observed in 8 times more reaction time in the same concentration. Eight times longer incubation time contributed to less than 2 times of log10 antiviral activity. The authors should clearly mention the bases of “time-dependence” of HIDROX inactivation activity using the obtained data in the present study.
L21: No need to keep “of action to show the utility of HIDROX” in this sentence.
L23: Please briefly describe the content of “cream”. Throughout the manuscript, the information related to “cream” is rarely mentioned.
L24-25: I cannot understand this sentence. I assume the author would like to argue the “intensity” of the bands of WB is weaker.
L53-57: I am so confused on the connection of these two sentences. Why the antiviral disinfectants are important to “compensate” the limitation of vaccines and treatments? I believe the antiviral disinfectants are playing the major role to reduce the risk of IAV circulation so as to raise the efficacy of vaccine and treatment “relatively”.
L82: Again, please specify the contents of “cream form”. The audience has no information.
L85-86: Please indicate the scientific bases that HIDROX may have potentials to show higher antiviral activity than HT; concentration of HT in HIDROX is higher than original HT, contain the molecule to indicate high antiviral activity, etc.
L102: I believe 7.25log10 should be expressed as 107.25.
L136: I propose not to describe the concentration of virus titer because the authors provided the information of purified virus solution at L102 already.
L149: Again, please specify “test cream”.
L151: I cannot understand why the 107.06 TCID50/60ul of unpurified virus is at higher concentration compared with the 107.25 TCID50/ml of purified virus. As well, the authors must mention the reason why the authors used the virus of 107.06 TCID50/60ul of virus for cream assay not using 107.25 TCID50/ml.
L196: same as the mu above comment, why the authors used the virus of 106.25 TCID50/ml of purified virus?
L207: Please specify “warrenoff reagent”.
Figure 1 and others: Why the authors indicate the figures in black-white format? I strongly suggest that the authors use more colors in the Figures (especially bar charts) to make them more easily to be visualized.
L287-288: The authors must put the information that this description was put without indication the statistic analysis. Otherwise, the authors may misunderstand the results.
Figure 2 and 3: these are incorrectly uploaded in the original manuscript. The authors must be more careful to finalize the manuscript.
L293: Why the detection limit(s) are different between the assays using 0.05 and 0.90 mg/ml of HT? The authors must explain this difference.
L295: It is not clear for me that these data was obtained by the calculation of virus titer from 12-16 independent mixture with single time TCID50 assay or from one mixture but carrying 12-16 TCID50 assay using the single same sample. Or, did the authors prepared 3 or 4 independent wells of mixtures, and carried out the TCID50 assay of each sample in triplicate? Please clarify this point.
L300: Please indicate the original concentration of HIDRAX in the cream forms so as to easily compare the antiviral activity in Results 3.2 and 3.3.
Figure 3: Please add the explanation of single star “*”; I assume this should be “p<0.05”.
Figure 3: According to the size of error bars, it is doubtful for me whether there is a significant difference between 0.90 ml/ml of HIDROX and HT at 3h.
L337-338: Was the number of ladder bands of HA, NA, M1 and NP at 24 h same? Please indicate in the main text.
L490-491: As far as I understand, the authors did not point out this in Results part. And, it is very hard for readers to agree this contents unless indicate the quantitative data of the band intensities.
Author Response
We are grateful to the reviewers for their valuable comments and helpful suggestions. We have revised the manuscript based on the reviewers’ comments. All the changes in the revised manuscript are highlighted in red color font. Further, we have provided our point-by-point responses to the reviewer’s comments below:
Reviewer #3:
Comment 1. Mohamed et al evaluated the inactivation activity of olive oil derived polyphenol against influenza A virus in vitro setting. The efficacy of HIDROX, one of the polyphenols evaluated in the present study seemed to be high. However, the authors should summarize the data more systemically to lead the authors more reasonable conclusions. Otherwise, so far in the current version of the manuscript, it must be hard to be published.
Response 1. We thank the reviewer for their insightful comment. We have conducted additional experiments and revised the manuscript to further support our findings.
Comment 2. Throughout the manuscript, I cannot get the idea what is possible mechanism of inactivation activity of HIDROX against IAV. HIDROX will alter the construction of the viral protein, but not limited to the outer membrane proteins but internal proteins. Potential change of outer membrane protein construction did not fully affect the hemagglutination and neuraminidase activities. How about the affect to polymerase activity? Again, the authors put the “potential” inactivation mechanisms of HIDROX against IAV, but those are rarely covered by the phenomenon obtained in the present study. In addition, the authors argued the HA2 could be the target of HIDROX without indicating the apparent evidences/bases of its hypothesis. This should be also critical to obtain the conclusion of the present study.
Response 2. We appreciate the reviewer’s comment. To further explore the mechanism of action of HIDROX and HT, we conducted additional experiments (Figure 9A, B). However, we could not obtain definitive data elucidating the precise mechanism of IAV inactivation by these test samples. Despite this limitation, we have addressed unresolved questions, study limitations, and potential hypotheses based on our findings (lines 626–628, 646–649, and 697–712). While viral polymerase activity in infected cells can be analyzed, assessing the effects of HIDROX and HT on polymerase activity within viral particles would be technically challenging and was not feasible in this study. Although this study does not fully clarify the virucidal mechanism of HIDROX and HT, we believe it offers valuable insights by revealing a previously uncharacterized aspect of naturally-derived virucidal compounds.
Comment 3. The authors may conclude that the antiviral activity of HIDROX was time-dependent manner. However, according to the results in Figure2. It might not be time-dependent manner. 1.46 log10 TCID50/ml decline was observed in 3h reaction of 0.05mg/ml of HT while only 2.34 log10 TCID50/ml decline was observed in 8 times more reaction time in the same concentration. Eight times longer incubation time contributed to less than 2 times of log10 antiviral activity. The authors should clearly mention the bases of “time-dependence” of HIDROX inactivation activity using the obtained data in the present study.
Response 3. Based on our prior experience in related studies, we observed that prolonging the reaction time does not always result in a proportional increase in virus inactivation efficiency. We therefore conclude that showing a strictly linear relationship between reaction time and virus inactivation efficiency is not essential to establish time dependency. The data show an apparent increase in virus inactivation efficiency between 3 h and 24 h. Given these findings, we determined that the virucidal activities of HIDROX and HT are time-dependent.
Comment 4. L21: No need to keep “of action to show the utility of HIDROX” in this sentence.
Response 4. We have addressed the reviewer’s comment by making the corresponding revision in line 20.
Comment 5. L23: Please briefly describe the content of “cream”. Throughout the manuscript, the information related to “cream” is rarely mentioned.
Response 5. We have added the cream composition details in lines 135–138.
Comment 6. L24-25: I cannot understand this sentence. I assume the author would like to argue the “intensity” of the bands of WB is weaker.
Response 6. We have revised the description in line 22.
Comment 7. L53-57: I am so confused on the connection of these two sentences. Why the antiviral disinfectants are important to “compensate” the limitation of vaccines and treatments? I believe the antiviral disinfectants are playing the major role to reduce the risk of IAV circulation so as to raise the efficacy of vaccine and treatment “relatively”.
Response 7. In response to the reviewer’s comment, we have revised the text in lines 57–60.
Comment 8. L82: Again, please specify the contents of “cream form”. The audience has no information.
Response 8. We have added the cream composition details in lines 135–138.
Comment 9. L85-86: Please indicate the scientific bases that HIDROX may have potentials to show higher antiviral activity than HT; concentration of HT in HIDROX is higher than original HT, contain the molecule to indicate high antiviral activity, etc.
Response 9. We apologize for not providing detailed information on the HIDROX and HT test concentrations. The concentration of HT in 0.90 mg/mL HIDROX is estimated to be 0.05 mg/mL. We have now included this explanation in lines 86–89. Accordingly, the tested HT concentrations were set at 0.90 and 0.05 mg/mL. By comparing 0.90 mg/mL HIDROX with these two HT concentrations, we aimed to determine whether the virus inactivation activity of HIDROX is attributable solely to HT or whether other components in HIDROX are also involved.
Comment 10. L102: I believe 7.25 log10 should be expressed as 107.25.
Response 10. The notation “x log10” has been replaced with “10x” throughout the manuscript, as suggested by the reviewer.
Comment 11. L136: I propose not to describe the concentration of virus titer because the authors provided the information of purified virus solution at L102 already.
Response 11. In line 116, we report the titer of the purified virus solution used to generate inactivated viruses. In line 156, we report the titer of the unpurified virus solutions. As these experiments were distinct, we documented the titer of each virus solution in the respective section.
Comment 12. L149: Again, please specify “test cream”.
Response 12. We have added the cream composition details in lines 135–138.
Comment 13. L151: I cannot understand why the 107.06 TCID50/60ul of unpurified virus is at higher concentration compared with the 107.25 TCID50/ml of purified virus. As well, the authors must mention the reason why the authors used the virus of 107.06 TCID50/60ul of virus for cream assay not using 107.25 TCID50/ml.
Response 13. We acknowledge that the viral titers used in Figures 1 and 2 differed. In preliminary cream-based experiments, we observed that when using the same titer as in the liquid-form experiments, the viruses became trapped in the base cream, resulting in unexpectedly low recovery even in the control group. This made it difficult to obtain reliable data. Based on these prior observations, we used a higher viral titer, as shown in Figure 2. However, the 0% HIDROX group yielded higher titers than anticipated. Due to limited remaining cream, we could not retest using the exact titer as in Figure 1. In the text (lines 570–574), we have clarified that the viral titers differed between Figures 1 and 2 and that direct comparisons between these results are invalid. While this is a study limitation, we explicitly state the viral titers used in each experiment to ensure readers can interpret the results accurately. The rationale for using 60 mL of viral solution in the cream experiment is provided in lines 182–183.
Comment 14. L196: same as the mu above comment, why the authors used the virus of 106.25 TCID50/ml of purified virus?
Response 14. To properly assess the neuraminidase (NA)-inhibition activities of HIDROX and HT, it was essential to establish an appropriate virus titer in the PBS control group. If the virus titer in the control group is too high, the experimental system may fail to function as intended. In the NA-inhibition assay—used to evaluate the inhibitory activity of antisera against the NA activity of IAV—the viral solution should be diluted to yield an optical density (OD) value of approximately 0.5, as recommended by the WHO Manual on Animal Influenza Diagnosis and Surveillance (https://www.chinacdc.cn/jkzt/crb/gjfd/zl/rgrgzbxqlg/jszl_2207/200510/P02005101124362428831469904020905865.pdf). Accordingly, we predetermined the virus titer (106.25 TCID50/mL) required to yield an OD value of approximately 0.5 in the PBS group for evaluating the NA-inhibition activities of HIDROX and HT. Similarly, a viral solution with an excessively high titer, containing large quantities of viral particles, is unsuitable for TEM observation. Therefore, for TEM (Figure 9A), we used a virus titer of 105.62 TCID50/mL, which allowed for the visualization of a reasonable number of virus particles. Since the optimal virus titer varies depending on the assay, we could not use the same viral titer across all experiments in this study.
Comment 15. L207: Please specify “warrenoff reagent”.
Response 15. We have provided additional details in lines 239–240.
Comment 16. Figure 1 and others: Why the authors indicate the figures in black-white format? I strongly suggest that the authors use more colors in the Figures (especially bar charts) to make them more easily to be visualized.
Response 16. As suggested by the reviewers, we have revised the figure to incorporate multiple colors.
Comment 17. L287-288: The authors must put the information that this description was put without indication the statistic analysis. Otherwise, the authors may misunderstand the results.
Response 17. In Figure 3, multiple comparison tests were conducted to evaluate the statistical differences across the four groups. Additional clarification has been provided in lines 360–362.
Comment 18. Figure 2 and 3: these are incorrectly uploaded in the original manuscript. The authors must be more careful to finalize the manuscript.
Response 18. We apologize for that error. We have corrected it in the revised manuscript.
Comment 19. L293: Why the detection limit(s) are different between the assays using 0.05 and 0.90 mg/ml of HT? The authors must explain this difference.
Response 19. We have provided additional explanation regarding cytotoxicity determination for each group in lines 165–170. However, the method for calculating the detection limit in each group is complex and has been excluded from the manuscript. The toxicity data for HIDROX and HT at each concentration in MDCK cells, obtained under limited experimental conditions, do not necessarily reflect toxicity in the human body. Furthermore, the process of determining the detection limit in each group is not directly relevant to the key findings of this study. To avoid unnecessarily complicating the manuscript, we have omitted the detailed calculation process for the detection limits.
Comment 20. L295: It is not clear for me that these data was obtained by the calculation of virus titer from 12-16 independent mixture with single time TCID50 assay or from one mixture but carrying 12-16 TCID50 assay using the single same sample. Or, did the authors prepared 3 or 4 independent wells of mixtures, and carried out the TCID50 assay of each sample in triplicate? Please clarify this point.
Response 20. For the 3-h reaction time experiment, we prepared four independent test tubes, each containing a mixture of virus solution and the respective test solution. After 3 h of incubation, the mixtures from all four tubes were inoculated into MDCK cells, and the virus titer of each mixture was determined. This experiment was repeated twice, resulting in 12 test tubes (n = 12 per group). The average virus titer was then calculated for each test group.
Comment 21. L300: Please indicate the original concentration of HIDRAX in the cream forms so as to easily compare the antiviral activity in Results 3.2 and 3.3.
Response 21. As stated in the manuscript, the HIDROX concentrations in the cream were 0%, 2%, 5%, and 10%. As noted in Response 13 to Reviewer #3, the virucidal activities of the HIDROX-containing solution and cream cannot be directly compared. This is a limitation of our study.
Comment 22. Figure 3: Please add the explanation of single star; I assume this should be “p<0.05”.
Response 22. We apologize for the incorrect placement of Figures 2 and 3 in the original manuscript. This error has now been corrected.
Comment 23. Figure 3: According to the size of error bars, it is doubtful for me whether there is a significant difference between 0.90 ml/ml of HIDROX and HT at 3h.
Response 23. We repeated the statistical analysis and confirmed the accuracy of the results.
Comment 24. L337-338: Was the number of ladder bands of HA, NA, M1 and NP at 24 h same? Please indicate in the main text.
Response 24. Determining the precise number of these bands appeared challenging.
Comment 25. L490-491: As far as I understand, the authors did not point out this in Results part. And, it is very hard for readers to agree this contents unless indicate the quantitative data of the band intensities.
Response 25. We agree with the reviewer’s comment and have removed the sentence from the revised manuscript.

Round 2
Reviewer 3 Report
Comments and Suggestions for Authors
The authors took adequate actions against most of my comments. However, the response against my commentsNO.3 is not enough. I am agreeing with the authors' final conclusion. But it is not obvious onto the manuscript.
The authors add the description why the authors consider time-dependent virucidal activity.
Author Response
We are grateful to the reviewer for his/her helpful suggestion. We have revised the manuscript based on the reviewer’s comment. All the changes in the revised manuscript are highlighted in red color font. Further, we have provided our response to the reviewer’s comment below:
Reviewer #3:
Comment. The authors took adequate actions against most of my comments. However, the response against my commentsNO.3 is not enough. I am agreeing with the authors' final conclusion. But it is not obvious onto the manuscript.
The authors add the description why the authors consider time-dependent virucidal activity.
Response. We have addressed the reviewer’s comment by adding the corresponding descriptions in lines 563–568.
